# Association between childhood immunisation coverage and proximity to health facilities in rural settings: a cross-sectional analysis of Service Provision Assessment 2013–2014 facility data and Demographic and Health Survey 2015–2016 individual data in Malawi

Nicole E Johns [1], Ahmad Reza Hosseinpoor [1], Mike Chisema [2]
M Carolina Danovaro-Holliday [3], Katherine Kirkby [1], Anne Schlotheuber [1],
Messeret Shibeshi [4], Samir V Sodha [3], Boston Zimba [5]

For numbered affiliations see end of article.

**Correspondence to**
Dr Ahmad Reza Hosseinpoor;
hosseinpoora@who.int

## ABSTRACT

**Objectives** Despite significant progress in childhood vaccination coverage globally, substantial inequality remains. Remote rural populations are recognised as a priority group for immunisation service equity. We aimed to link facility and individual data to examine the relationship between distance to services and immunisation coverage empirically, specifically using a rural population.

**Design and setting** Retrospective cross-sectional analysis of facility data from the 2013–2014 Malawi Service Provision Assessment and individual data from the 2015–2016 Malawi Demographic and Health Survey, linking children to facilities within a 5 km radius. We examined associations between proximity to health facilities and vaccination receipt via bivariate comparisons and logistic regression models.

**Participants** 2740 children aged 12–23 months living in rural areas.

**Outcome measures** Immunisation coverage for the six vaccines included in the Malawi Expanded Programme on Immunization schedule for children under 1 year at time of study, as well as two composite vaccination indicators (receipt of basic vaccines and receipt of all recommended vaccines), zero-dose pentavalent coverage, and pentavalent dropout.

**Findings** 72% (706/977) of facilities offered childhood vaccination services. Among children in rural areas, 61% were proximal to (within 5 km of) a vaccine-providing facility. Proximity to a vaccine-providing health facility was associated with increased likelihood of having received the rotavirus vaccine (93% vs 88%, p=0.004) and measles vaccine (93% vs 89%, p=0.01) in bivariate tests. In adjusted comparisons, how close a child was to a health facility remained meaningfully associated with how likely they were to have received rotavirus vaccine (adjusted

## STRENGTHS AND LIMITATIONS OF THIS STUDY

⇒ Health facility and individual data were linked to examine immunisation service access and equality for rural populations using a nationally representative dataset.

⇒ Straight-line distance to nearest immunisation service was calculated, and the association between proximity to a facility providing services and child-level immunisation coverage outcomes were examined.

⇒ Analyses accounted for additional known predictors of immunisation coverage such as birth order and household wealth.

⇒ Straight-line distance to nearest services does not reflect actual access, which is tied to road networks, geography, seasonality, vehicle access and preferred facilities.

⇒ Distance is only one of many factors which render a population hard-to-reach or hard-to-vaccinate.

OR (AOR) 1.63, 95% CI 1.13 to 2.33) and measles vaccine (AOR 1.62, 95% CI 1.11 to 2.37).

**Conclusion** Proximity to health facilities was significantly associated with likelihood of receipt for some, but not all, vaccines. Our findings reiterate the vulnerability of children residing far from static vaccination services; efforts that specifically target remote rural populations living far from health facilities are warranted to ensure equitable vaccination coverage.

## INTRODUCTION

Despite significant progress in childhood vaccination coverage globally since the

establishment of the Expanded Programme on Immunization (EPI) in 1974, substantial inequality in vaccination coverage remains.[1–4] Within countries, immunisation coverage tends to be lowest among the most disadvantaged subgroups, including those living in rural areas.[2 5 6] The Equity Reference Group for Immunization has identified remote rural populations (those living furthest from population centres) as one of the most pressing priority areas of work for improving immunisation equity.[7 8] The Immunization Agenda 2030 (IA2030) recognises coverage and equity as strategic priorities and has the stated goal to leave no one behind, that is, to ensure that hard-to-reach and marginalised populations are included and centred in immunisation policies and initiatives, and to ultimately achieve universal immunisation coverage.[9 10] Similarly, Gavi's most recent strategy also centres the theme of leaving no one behind and indicates equity as an organising principle and key goal.[11] Globally, governments and key stakeholders in immunisation are recognising equity, including reaching remote rural populations, as a central priority.[8]

Immunisation services for remote rural populations involve a number of barriers. On the side of the procurement, distribution and provision of vaccines, barriers include increased marginal costs of administration, challenges of sufficient health worker availability and training, physical access for supply delivery, and cold-chain continuity.[7] One of the largest barriers on the side of the remote rural patient to immunisation provision is physical access to static or outreach services. Distance to a health facility is a commonly cited barrier to vaccination in nationally representative studies and systematic reviews.[12–15] Several studies have examined distance empirically, and find significant negative associations between distance to vaccination site and likelihood of vaccination, though all rely on subnational samples.[16–18] One nationally representative study from Nigeria found that distance to a health facility (whether it provided vaccination services or not) was significantly and negatively associated with receipt of most vaccines.[19] Distance to a health facility that routinely provides immunisation services is likely associated with immunisation coverage, particularly among rural populations.

The Malawi health system consists of public, private for profit and private not-for-profit sectors; the majority of healthcare services are provided by the government, followed by the major religious provider Christian Health Association of Malawi (CHAM).[20] Though vaccination services may be available at all facilities, the majority of children receive vaccinations through government facilities or outreach clinics.[21] In the community, immunisation services are most frequently provided by health workers known as health surveillance assistants, who provide door-to-door visitations and staff village, outreach and mobile clinics.[20 22] Routine immunisation services are also supported by other periodic supplementary immunisation activities.[23–25]

As of the 2015–2016 Malawi Demographic and Health Survey (DHS), an estimated 76% (95% CI 74% to 78%) of children aged 12–23 months had received basic immunisations (defined as one dose of BCG; three doses of oral polio vaccine (OPV); three doses of pentavalent vaccine (diphtheria, tetanus, pertussis, hepatitis B virus and *Haemophilus influenzae* type b (Hib)); and one dose of measles vaccine), a decline from 81% (95% CI 79% to 83%) in the 2010–2011 DHS.[26] Contrary to findings in other sub-Saharan settings and prior research within Malawi, childhood vaccination coverage was in fact higher among children living in rural areas (77%, 95% CI 74% to 79%) than children living in urban areas (70%, 95% CI 63% to 76%).[12 26] However, such figures conceal inequalities in access to immunisation *within* rural populations, where the most remote are still likely under-immunised.[7] Data from the 2015–2016 Malawi DHS confirms these within-rural inequalities: coverage of basic vaccines varied by more than 25 percentage points across rural strata, from 65% (95% CI 53% to 75%) to 91% (95% CI 84% to 95%).[27] Understanding inequities within rural populations is particularly relevant in Malawi, where 84% of the population lives in rural areas.[28]

The policy of the Malawi Ministry of Health is that all Malawians should live within 8 km of a public health facility, though as of 2014 more than 2 million people were estimated to live further than this from care.[29] The 8 km policy is intended to ensure 'reasonable walking distance' to healthcare, which is particularly relevant given low vehicle ownership (only 4% of households in rural areas had a car, truck, motorcycle or scooter as of the 2015–2016 DHS).[27] Several studies from Malawi suggest that residing far from a facility providing vaccination services is a driver of inequitable immunisation coverage for remote rural populations.[30 31] Research has documented the relationship between empirically derived distance to facility and other healthcare services and outcomes in Malawi;[32–37] to our knowledge, no other study has examined empirically derived distance to facilities providing vaccination services and immunisation coverage in Malawi.

In this study, we hypothesise that children living in rural areas aged 12–23 months who are proximal to a vaccine-providing health facility will be significantly more likely to have received immunisations than children who live further than 5 km from the nearest vaccine-providing facility. Substantial research has explored individual-level factors associated with vaccination, but less work has demonstrated an association between empirically derived facility access and immunisation coverage, particularly using nationally representative data. The nature of the data available in Malawi poses a unique opportunity to explore this quantitatively.

## METHODS
### Data sources
Facility data come from the Malawi 2013–2014 DHS Service Provision Assessment (SPA) survey, a census of formal-sector public and private health facilities in Malawi.[38] It includes all static sites but not outreach clinic

**Table 1** Malawi EPI vaccination schedule for children under 1 year, 2015*

| Age | Vaccine |
| --- | --- |
| Birth up to 2 weeks after birth | BCG<br>Birth dose OPV |
| 6 weeks | OPV dose 1<br>Pentavalent dose 1<br>PCV13 dose 1<br>Rotavirus dose 1 |
| 10 weeks | OPV dose 2<br>Pentavalent dose 2<br>PCV13 dose 2<br>Rotavirus dose 2 |
| 14 weeks | OPV dose 3<br>Pentavalent dose 3<br>PCV13 dose 3 |
| 9 months | Measles dose 1 |

*Though this schedule reflects the EPI schedule at time of data analysed here, several changes to the EPI have gone into effect since 2015. The measles vaccine was replaced with measles and rubella vaccine in July 2017. The trivalent OPV was replaced with bivalent OPV in 2016, and inactivated polio vaccine (IPV) was introduced starting in December 2018. The malaria vaccine (RTS,S/ASO1) was also piloted in 2019 but is not yet in the national EPI vaccination schedule as of May 2022.
EPI, Expanded Programme on Immunization; OPV, oral polio vaccine; PCV13, pneumococcal conjugate vaccine 13-valent.

locations. Facilities included in the survey were hospitals, health centres, dispensaries, clinics and health posts; managing authorities included government, CHAM and other faith-based organisations, private sector for-profit organisations, non-governmental organisations (NGOs) and companies.

We used data from the Malawi 2015–2016 DHS, the most recent available DHS survey in the country and the closest in time to the SPA, to determine immunisation coverage among children aged 12–23 months residing in rural areas. Vaccine doses included were those six in the Malawi EPI schedule for children under 1 year at the time of data collection: BCG, OPV, pentavalent, pneumococcal conjugate vaccine 13-valent (PCV13), rotavirus vaccine and measles-containing vaccine (table 1).[39] Details regarding DHS sampling, implementation and content are published elsewhere.[26] Individual, family and community characteristics were also extracted from this dataset.

### Geographic data linkage
Facility geolocation, provided as part of SPA data, was matched to individual data using the DHS-suggested technique of a Euclidian (straight-line) distance buffer around each DHS cluster centroid.[40] This buffer is defined as 5 km in rural areas; this corresponds with the DHS offset of cluster GPS locations by up to 5 km in rural areas. Therefore, we defined proximity to a vaccine-providing facility as children living in rural areas within 5 km of a facility that reported regularly providing vaccination services on-site and/or via outreach. We also tested alternate buffer distances, including 8 km based on Malawi Ministry of Health definitions (see sensitivity analysis description later).[29]

### Measures
Facility data indicated vaccination provision, stock, fees and preparedness. For the sake of these analyses, only presence of a health facility offering vaccination services was used. Indicators of vaccination fees were not used because very few facilities (2%) reported charging vaccination-specific fees. Indicators of full EPI provision were not used because most facilities offering any vaccination services offered all EPI vaccines (99% of facilities offering vaccines offered full EPI). Despite a 2–3 years gap between the survey of facilities (2013–2014) and survey of individuals (2015–2016, children born October 2013–February 2015), retroactive reporting of immunisation for children 12–23 months reflects immunisation receipt close to the time of facility survey, and provision of any vaccination services is likely to have remained constant over that time.

Immunisation coverage was defined as the proportion of children aged 12–23 months who received the indicated vaccine dose by the time of survey. For OPV, pentavalent and PCV13 vaccines, children were considered having received the complete series of each vaccine type if they had three doses. Birth dose OPV was considered separately from the three-dose OPV series. For rotavirus vaccine, two doses were considered the complete series, and for BCG and measles, one dose (or more) was considered complete. We also examined receipt of basic vaccines, also termed full immunisation with basic vaccine doses (one dose BCG, three doses OPV, three doses pentavalent and one dose measles) as well as receipt of all recommended vaccines (basic vaccine doses plus birth dose OPV, two doses rotavirus and three doses PCV13). Finally, we created indicators of zero-dose pentavalent coverage (no receipt of pentavalent vaccine) and pentavalent dropout (receipt of the first dose, but not the third dose, of pentavalent vaccine). For all vaccines, both mother's recall and vaccination card verification (with or without date) were used to determine coverage. Similarly, vaccines received within the recommended time frame and any time after were included. The few 'don't know' responses were coded as non-receipt of the corresponding vaccine/dose (<1% for all, N=0–10 per vaccine dose).

### Analyses
All analyses were limited to rural populations, as defined by DHS.[41] We first used bivariate Pearson's $\chi^2$ tests to assess associations between vaccination coverage and proximity to a vaccine-providing health facility among children living in rural areas, for all immunisation outcomes. We then constructed unadjusted and adjusted logistic regression models to assess the association between proximity to a vaccine-providing health facility and immunisation

outcomes among children living in rural areas, with adjusted models controlling for: child's sex (male, female), birth order (1, 2, 3, 4, 5+), household wealth (quintile), mother's age (years), mother's education (none, primary, secondary or higher) and subnational region (Northern, Central, Southern). All covariates were selected a priori based on established association with likelihood of immunisation and availability in DHS survey data.[21 31 42]

We then conducted several sensitivity analyses. First, we constructed multinomial models to separate immunisation verified by vaccine card and immunisation noted via mother's recall, to examine whether the association between facility proximity and immunisation was sensitive to the nature of vaccination ascertainment. For each vaccine, we defined a 3-level outcome as: not received (referent group), received and verified by vaccine card, and received as noted via mother's recall. Second, we modified to the proximity radius to 8 km rather than the 5 km suggested by DHS, to align with Malawi Ministry of Health policy and assess whether findings were sensitive to the distance radius used.[29 40] Third, we restricted the sample to children residing within 5 km of a vaccine-providing facility, and examined whether facility level of care and managing authority were associated with vaccination coverage in adjusted logistic regression models. We captured facility level of care via a single variable, classified as: hospital, health centre or health post/clinic/dispensary. We assigned children who were within 5 km of multiple facilities providing vaccines to the highest level of care among those proximal facilities. We captured facility managing authority via a set of four indicators, separately assessing whether a child was proximal to a vaccine-providing facility managed by the government, CHAM, private for-profit and/or NGO. Fourth, as a final post-hoc analysis, we examined vaccine stock at time of SPA survey, considering a vaccine in stock if it was observed, or reported to be available, and was not expired. For each of the six examined vaccines, we replicated adjusted analyses of immunisation coverage for the total sample using an indicator of the corresponding vaccine stock within 5 km, and the same analyses limited to children within 5 km of a vaccine-providing facility.

All individual-level analyses accounted for complex sampling design using provided DHS survey weights. Statistical significance was set at $p<0.05$ for Pearson's $\chi^2$ tests, adjusted ORs (AORs) and adjusted relative risk ratios; 95% CIs are reported throughout. We conducted analyses using ArcGIS Pro V.2.8.0 and STATA V.15.1.

### Patient and public involvement
Due to the nature of this analysis, patients and the public were not involved in the design, conduct, reporting, or dissemination of this research.

### RESULTS
Of 977 facilities surveyed, 72% (N=706) offered any childhood immunisation services (table 2).

**Table 2** Health facilities offering childhood immunisation services, Malawi SPA 2013–2014

|  | N | % |
|---|---|---|
| Total | 706 | 100.0 |
| Type of vaccination offerings |  |  |
| % offering all EPI vaccines | 696 | 98.6 |
| Facility type |  |  |
| Central hospital | 1 | 0.1 |
| District hospital | 24 | 3.4 |
| Rural/community hospital | 41 | 5.8 |
| Other hospital | 31 | 4.4 |
| Health centre | 465 | 65.9 |
| Maternity | 3 | 0.4 |
| Dispensary | 40 | 5.7 |
| Clinic | 81 | 11.5 |
| Health post | 20 | 2.8 |
| Managing authority |  |  |
| Government/public | 457 | 64.7 |
| Christian Health Association of Malawi (CHAM) | 153 | 21.7 |
| Private for profit | 39 | 5.5 |
| Other mission/faith-based | 6 | 0.9 |
| Non-governmental organisation (NGO) | 19 | 2.7 |
| Company | 32 | 4.5 |

EPI, Expanded Programme on Immunization; SPA, Service Provision Assessment.

Individual survey data included 2740 children aged 12–23 months living in rural areas (online supplemental appendix table 1). All living children aged 12–23 months residing in rural areas were included. The majority of children were proximal to health facilities: 64% were within 5 km of *any* health facility and 61% were within 5 km of a vaccine-providing facility (table 3). Proximity was often limited to a single facility; only 5% were within 5 km of two or more vaccine-providing facilities.

Overall, immunisation coverage among children in rural areas was above 80% for all examined individual vaccines: 97.5% 1-dose BCG, 82.2% 3-dose OPV, 93.4% 3-dose pentavalent, 91.1% 2-dose rotavirus, 89.7% 3-dose PCV13 and 91.7% 1-dose measles (table 3). However, full immunisation coverage was lower: three-fourths (76.8%) of children had basic vaccine doses and only half (50.4%) had all vaccines recommended for children under 1 year. Few children received no doses of pentavalent vaccine (2.5%), and among children who started their pentavalent vaccine series, 4.2% did not complete it (indicating dropout). Among children with vaccination date information available, all vaccines except measles had 99% or greater receipt within the first year of life, while 93.4% of measles first doses were received before age one (results not shown).

**Table 3** Study sample—children aged 12–23 months in rural areas, Malawi DHS 2015–2016

| | Unweighted N | Weighted % |
|---|---|---|
| Total | 2740 | 100.0 |
| Facility proximity indicators | | |
| Facility proximity within 5 km | | |
| No facility within 5 km | 978 | 36.3 |
| Any facility within 5 km | 1762 | 63.7 |
| Vaccine-providing facility within 5 km | 1698 | 61.2 |
| Non-vaccine-providing facility only within 5 km | 64 | 2.6 |
| Proximal to vaccine-providing facility, by type* | | |
| Hospital | 403 | 12.1 |
| Health centre | 1264 | 47.2 |
| Health post/clinic/ dispensary | 289 | 10.9 |
| Proximal to vaccine-providing facility, by managing authority* | | |
| Government/public | 1305 | 46.7 |
| CHAM | 501 | 16.9 |
| Private for profit | 53 | 2.8 |
| NGO | 74 | 2.8 |
| Immunisation coverage indicators | | |
| Specific immunisation dose coverage | | |
| BCG | 2673 | 97.5 |
| OPV 3 doses | 2273 | 82.2 |
| Pentavalent 3 doses | 2559 | 93.4 |
| Rotavirus 2 doses | 2510 | 91.1 |
| PCV13 3 doses | 2460 | 89.7 |
| Measles 1+ dose | 2524 | 91.7 |
| Coverage of group of immunisations | | |
| Basic vaccines† | 2129 | 76.8 |
| All recommended vaccines‡ | 1450 | 50.4 |
| Negative immunisation outcomes | | |
| Pentavalent zero dose | 61 | 2.5 |
| Pentavalent dropout§ | 120 | 4.2 |

*Children could be proximal to more than one vaccine-providing facility and thus more than one vaccine-providing facility type/ managing authority.
†Defined as BCG one dose, OPV three doses, diphtheria, tetanus, pertussis (DTP)/hepatitis B virus (HBV)/*Haemophilus influenzae* type b (Hib) (pentavalent) three doses, measles one dose.
‡Basic+OPV birth dose+rotavirus two doses+PCV13 three doses.
§Dropout denominator is children with at least one dose of pentavalent; represents per cent of children receiving first but not third dose of pentavalent.
CHAM, Christian Health Association of Malawi; DHS, Demographic and Health Survey; NGO, non-governmental organisation; OPV, oral polio vaccine; PCV13, pneumococcal conjugate vaccine 13-valent.

Children living in rural areas who were proximal to a vaccine-providing facility were more likely to have some, but not all, vaccine doses in unadjusted bivariate tests when compared with children who were not in proximity to a vaccine-providing facility: rotavirus (93% vs 88%, p=0.004), measles (93% vs 89%, p=0.01) and full immunisation with basic vaccines (79% vs 74%, p=0.04) and with all recommended vaccines (54% vs 45%, p<0.001) (table 4). No significant bivariate differences were observed for BCG, OPV, pentavalent, or PCV13 vaccine, nor pentavalent zero dose or dropout.

These relationships hold true in fully adjusted regression models (table 4). Compared with children living greater than 5 km from the nearest facility providing immunisation services, children living proximal to a vaccine-providing facility had greater odds of receiving rotavirus (AOR 1.63, 95% CI 1.13 to 2.33) and measles (AOR 1.62, 95% CI 1.11 to 2.37) vaccines. They also had marginally greater odds of receiving basic vaccines (AOR 1.28, 95% CI 0.99 to 1.65, p=0.057) and greater odds of receiving all recommended vaccines (AOR 1.38, 95% CI 1.09 to 1.74). They had somewhat lower odds of being zero-dose for pentavalent (AOR 0.53, 95% CI 0.28 to 1.01, p=0.052). No statistically significant associations were observed with BCG, OPV, pentavalent, or PCV13 receipt, nor pentavalent dropout.

Associations between proximity to a vaccine-providing facility and immunisation coverage were similar for both mother's recall and vaccination card documented immunisation in multinomial models (table 4). The majority of children in rural areas (80%, 95% CI 78% to 82%) had a vaccination card which was seen. Children were both more likely to have vaccination card-recorded rotavirus vaccination (AOR 1.71, 95% CI 1.12 to 2.60) or mother-recalled rotavirus vaccination (AOR 1.72, 95% CI 1.18 to 2.51) if they were proximal to a vaccine-providing facility. Children were also both more likely to have vaccination card recorded measles vaccination (AOR 1.59, 95% CI 1.07 to 2.35) or mother-recalled measles vaccination (AOR 1.52, 95% CI 1.04 to 2.24) if they were proximal to a vaccine-providing facility. No significant associations between proximity to vaccine-providing facility and immunisation, whether ascertained through mother recall or vaccination card, were observed for BCG, OPV, pentavalent or PCV13 vaccines.

When considering proximity as 8 km rather than 5 km, 87% of children were proximal to a vaccine-providing facility; associations with immunisation coverage were similar to those with 5 km proximity definition (online supplemental appendix table 2). We observed significantly greater immunisation coverage among children within 8 km of a vaccine-providing facility for rotavirus (91.9% vs 87.0%, p=0.02), measles (92.0% vs 86.0%, p=0.03) and all recommended vaccines (53.2% vs 36.7%, p<0.001) in unadjusted comparisons. Additionally, zero-dose pentavalent receipt was significantly less common among children within 8 km of a vaccine-providing facility compared with those not (2.3% vs 5.4%, p=0.03). Findings were similar

**Table 4** Unadjusted and adjusted comparisons of immunisation rates among children in rural areas aged 12–23 months by presence of vaccine-providing facility in 5 km radius, Malawi 2013–2014

| | Unadjusted rates | | P value | Unadjusted logistic regression | | Adjusted logistic regression | | Adjusted multinomial logistic regression, three-level recording-method-specific outcome | | | |
| --- | --- | --- | --- | --- | --- | --- | --- | --- | --- | --- | --- |
| | | | | | | | | Mother recall | | Vaccine card | |
| | No vaccine-providing facility proximal (%) | Vaccine-providing facility proximal (%) | | OR† | 95% CI | AOR†‡ | 95% CI | aRRR†‡ | 95% CI | aRRR†‡ | 95% CI |
| **Specific immunisation dose coverage** | | | | | | | | | | | |
| BCG | 97.4 | 97.6 | 0.78 | 1.09 | (0.58 to 2.05) | 1.17 | (0.63 to 2.20) | 1.28 | (0.67 to 2.42) | 1.15 | (0.61 to 2.16) |
| Rotavirus two doses | 88.5 | 92.8 | 0.004 | 1.68** | (1.17 to 2.41) | 1.63** | (1.13 to 2.33) | 1.71* | (1.12 to 2.60) | 1.72** | (1.18 to 2.51) |
| OPV three doses | 81.4 | 82.6 | 0.55 | 1.09 | (0.83 to 1.42) | 1.08 | (0.83 to 1.40) | 1.15 | (0.75 to 1.75) | 1.11 | (0.85 to 1.46) |
| Pentavalent three doses | 92.6 | 94.0 | 0.30 | 1.24 | (0.83 to 1.87) | 1.23 | (0.84 to 1.82) | 1.33 | (0.85 to 2.09) | 1.20 | (0.80 to 1.79) |
| PCV13 three doses | 88.9 | 90.2 | 0.49 | 1.15 | (0.78 to 1.68) | 1.07 | (0.75 to 1.53) | 1.09 | (0.70 to 1.68) | 1.10 | (0.76 to 1.59) |
| Measles 1+ dose | 89.3 | 93.3 | 0.01 | 1.68** | (1.14 to 2.48) | 1.62* | (1.11 to 2.37) | 1.52* | (1.04 to 2.24) | 1.59* | (1.07 to 2.35) |
| **Coverage of group of immunisations** | | | | | | | | | | | |
| Basic vaccines§ | 73.8 | 78.7 | 0.04 | 1.31* | (1.01 to 1.69) | 1.28 | (0.99 to 1.65) | – | – | – | – |
| All recommended vaccines¶ | 45.0 | 53.7 | <0.001 | 1.42** | (1.13 to 1.78) | 1.38** | (1.09 to 1.74) | – | – | – | – |
| **Negative immunisation outcomes** | | | | | | | | | | | |
| Pentavalent zero dose | 3.4 | 1.9 | 0.08 | 0.55 | (0.28 to 1.10) | 0.53 | (0.28 to 1.01) | – | – | – | – |
| Pentavalent dropout†† | 4.1 | 4.2 | 0.91 | 1.02 | (0.67 to 1.56) | 1.05 | (0.69 to 1.61) | – | – | – | – |

*p<0.05, **p<0.01, ***p<0.001.
†Reference is children not proximal to a vaccine-providing facility.
‡Models also control for household wealth, mother's education, mother's age, child's sex, child's birth order, region of country.
§Defined as BCG one dose, OPV three doses, diphtheria, tetanus, pertussis (DTP)/hepatitis B virus (HBV)/*Haemophilus influenzae* type b (Hib) (pentavalent) three doses, measles one dose.
¶Basic+OPV birth dose+rotavirus two doses+PCV13 three doses.
††Dropout denominator is children with at least one dose of pentavalent.
AOR, adjusted OR; aRRR, adjusted relative risk ratio; OPV, oral polio vaccine; PCV13, pneumococcal conjugate vaccine 13-valent.

in adjusted models: measles (AOR 1.88, 95% CI 1.06 to 3.33, p=0.03), basic vaccines (AOR 1.54, 95% CI 1.02 to 2.32, p=0.04), all recommended vaccines (AOR 1.97, 95% CI 1.35 to 2.89, p<0.001), zero-dose pentavalent (AOR 0.38, 95% CI 0.18 to 0.83, p=0.01). Rotavirus vaccine did not have a significant association with 8 km proximity to a vaccine-providing facility in adjusted models (AOR 1.53, 95% CI 0.95 to 2.45, p=0.08). As with 5 km findings, no significant associations were observed for OPV, BCG, pentavalent, or PCV13 vaccines.

When limited to children within 5 km of a vaccine-providing facility, the level of the facility was not associated with any of the immunisation indicators (online supplemental appendix table 3A,B). Proximity to a government-run facility offering vaccination services was associated with greater odds of receipt of 3-dose OPV (AOR 2.32, 95% CI 1.37 to 3.93), basic vaccines (AOR 1.82, 95% CI 1.15 to 2.88), and all recommended vaccines (AOR 1.94, 95% CI 1.24 to 3.03), and lower odds of zero-dose pentavalent (AOR 0.08, 95% CI 0.02 to 0.33). Proximity to a CHAM facility offering vaccination services was associated with greater odds of receipt of 3-dose OPV (AOR 2.47, 95% CI 1.46 to 4.18), 3-dose pentavalent (AOR 2.04, 95% CI 1.06 to 3.93), and basic vaccines (AOR 2.29, 95% CI 1.44 to 3.63), and lower odds of zero-dose pentavalent (AOR 0.11, 95% CI 0.02 to 0.50).

Current vaccine stock was positively and significantly associated with rotavirus (AOR 1.67, 95% CI 1.17 to 2.39) and measles (AOR 1.52, 95% CI 1.04 to 2.22) immunisation coverage; these findings were consistent to those using indicators of immunisation service availability generally (results not shown). When limited to children living within 5 km of a vaccine-providing facility, vaccine stock was not associated with immunisation coverage for any of the six examined vaccines.

## DISCUSSION

Among children aged 12–23 months living in rural areas, proximity to a health facility providing vaccination services was associated with increased likelihood of rotavirus and measles vaccine receipt (and therefore receipt of all recommended vaccines, as rotavirus and measles are part of this composite indicator), as well as with decreased likelihood of zero-dose pentavalent vaccination. Even when accounting for known child, mother, family and geographic determinants of immunisation, how close a child was to a health facility was meaningfully associated with how likely they were to have received certain vaccines.

Proximity to a vaccine-providing health facility among rural children can be considered a proxy for remote rural-ness: those who are far from a vaccine-providing facility are also likely to be far from a population centre and far from services more generally. Therefore, these findings indicate that remote rural children in Malawi were likely inequitably under-vaccinated at the time of the 2015–2016 DHS. This is an important equity consideration;

despite higher immunisation coverage among rural children than urban children overall, these findings reiterate that rural populations are not a monolith and that inequities are present beyond urban/rural differences. While geographic distance is just one of many factors which render a population hard-to-reach or hard-to-vaccinate, it is relatively easy to define and target, and is a meaningful correlate of coverage.[43] Equity-focused interventions and monitoring efforts should therefore use as granular geographic delineations as possible, with particular focus on those populations furthest from care.

We observed differential associations by type of immunisation. The measles vaccine is delivered on its own several months after other infant vaccinations, and the requirement of a specific healthcare visit to obtain it may exacerbate the barrier of increased distance for accessing care. While rotavirus vaccine is offered simultaneously with other vaccinations, it was introduced to the Malawi EPI in October 2012 and outreach efforts may still have been in scale-up at the time the children under consideration were eligible for vaccination. It was introduced with strict age restrictions (first dose at 6–12 weeks, 4 weeks between doses, second dose no later than 16 weeks), which were not formally removed from the Malawi EPI until 2017.[44] The narrow time range for vaccination created by these limits may have made the vaccine harder to access for populations far from facilities. Conversely, the null findings for 3-dose OPV and 3-dose pentavalent vaccine add evidence to the success of the Malawi EPI in ensuring access to these vaccines among rural populations more broadly.[39]

In analyses restricted to children living within 5 km of vaccine-providing facilities, we observed no differences in immunisation coverage by facility level of care, but we did find significant differences by facility managing authority. Similar odds of immunisation regardless of facility-level are unsurprising given that immunisation services do not require highly specialised equipment or intensive provider training. Greater likelihood of immunisation when the proximal facility was managed by the government or CHAM likely reflects the higher rate of outreach efforts performed by these authorities, the lack of fees for immunisation at these facilities, and greater resource availability, training, and oversight more broadly.[45] Additional public-sector outreach efforts in areas where the only health facilities are run privately may thus be warranted.

Our finding that immunisation coverage is inequitably lower among children further from health facilities is particularly relevant in light of the COVID-19 pandemic and its effects on childhood immunisation services. Pandemic effects on childhood immunisation services include service provision limitations such as suspension of outreach activities, disruption and suspension of in-facility services, disruption to vaccine and supply availability, shortages of available healthcare workers and service utilisation limitations including travel restrictions, concern for health and safety in seeking services,

and lack of knowledge of service availability.[46 47] Mitigation efforts within Malawi and in other country contexts reduced the disruptions to routine immunisation, resulting in only a 1%–2% decline in coverage of vaccinations at the national level for 2020.[48–50] However, remote rural patients are most likely to have experienced these disruptions, given the reliance on outreach services or on travel to seek care. Strategies will be needed to ensure that missed children are caught-up for equitable immunisation coverage.[51] Practically, these additional efforts should include campaigns and outreach efforts, as these are less resource-intensive to implement than the construction, staffing, and maintenance of new facilities. These outreach efforts can be tailored to reach the most rural populations by inclusion of transportation considerations such as providing cars, motor bikes, and fuel, as well as supplies which can be carried long distances and be used in areas with limited infrastructure. Our findings also add further support to the stated goal of the Malawi Ministry of Health that all Malawians live within 8 km of a health facility, and the construction of additional facilities should continue to prioritise those areas where people are furthest from care.

### Limitations

These findings must be considered in light of several limitations. First, individual data does not indicate where vaccination services were rendered; children may have received immunisation services from an alternate facility than the one most proximal to them (including via outreach, the locations of which were not assessed in the SPA). Thus, it would be inappropriate to suggest a causal relationship between the most proximal facility's immunisation service availability and an individual child's immunisation. Additionally, outreach services are widely used in this population, with more than 5000 fixed and mobile outreach clinics throughout the country[39]; the observed association of immunisation coverage with distance to static clinics likely underestimates the true strength of association with distance to location where vaccination was actually received. Second, general service availability may not reflect actual service readiness and availability at the time of immunisation receipt. Third, three planned sensitivity analyses were not possible: separately analysing facilities offering within-facility and outreach services was not possible as 99% of children who were proximal to a vaccine-providing facility were proximal to one offering both in-facility and outreach services; examining specific vaccination availability was not possible because 99% of children who were proximal to a vaccine-providing facility were proximal to one offering all six examined vaccines; examining receipt of vaccines within vs after the first year was not possible because >99% of children who had vaccination dates available received all vaccines by age one (with the exception of measles vaccine, which had 93% receipt by age one). Fourth, while the most recent available surveys were used, at the time of publication these data are now 6–9 years old; additional research

using more recent data will add insight into current realities. Furthermore, there was a 2–3 years gap between the survey of facilities (2013–2014) and survey of individuals (2015–2016, children born October 2013–February 2015), and facilities may have closed, opened, changed service offerings, or had fluctuations in vaccine stock over that time frame. However, retroactive reporting of immunisation reflects immunisation receipt close to the time of facility survey, and provision of any vaccination services is likely to have remained constant over that time. The lack of association between vaccine stock and immunisation coverage may be due in part to the asynchronous surveys. Finally, the use of buffer distances does not account for actual travel distance, geography, or time, nor does it account for seasonal differences in physical access (eg, due to rains). Similarly, the use of DHS cluster centroids does not reflect the actual household geolocation. Statistically, this will bias findings toward the null as the buffer distance used will not be precise; any associations observed are likely underestimates of the true strength of association. Recent studies support the use of other methods such as theoretical catchment areas for more accurate facility linkage.[52] However, given the research question and the variable size of administrative and catchment areas, buffer distance was considered appropriate for these analyses.

### CONCLUSION

Findings from this study align with previous work demonstrating a significant association between immunisation coverage and distance to vaccine-providing facilities,[16–19] and expand on this by using a nationally representative sample with a focus on children living in rural areas. Remote rural populations have been identified as a key target for improving immunisation equity, and these findings reiterate the vulnerability of children residing far from static vaccination services. Efforts that target remote rural populations living far from health facilities, even using crude measures of identification such as straight-line distance from facilities, are warranted to ensure equitable vaccination coverage. These analyses also suggest that health facility-level data can and should be used for further analyses of inequalities in immunisation.

**Author affiliations**
[1]Department of Data and Analytics, World Health Organization, Geneve, Switzerland
[2]Preventive Health Services and Expanded Program on Immunization, Malawi Ministry of Health, Lilongwe, Malawi
[3]Department of Immunization, Vaccines and Biologicals, World Health Organization, Geneve, Switzerland
[4]Inter-Country Support Team for East and Southern Africa, World Health Organization, Harare, Zimbabwe
[5]Malawi Country Office, World Health Organization, Lilongwe, Malawi

**Contributors** ARH led initial study conceptualisation and provided oversight of analyses and writing. NEJ generated specific hypotheses, led data analyses and created initial manuscript draft. MC, MS and BZ provided country-specific context, content and feedback for interpretation of the data. MCD-H and SVS provided immunisation context, content and feedback for interpretation of the data. KK

and AS provided health equity context, content and feedback for interpretation of the data. All authors contributed to substantive review and revision of the final manuscript, provided final approval of the version to be published and agree to be accountable for all aspects of the work. ARH is the guarantor and accepts full responsibility for the finished work, had access to the data, and controlled the decision to publish.

**Funding** Funding for this work was provided by Gavi, the Vaccine Alliance. The funding source had no role in the study design, analysis, interpretation of data or decision to publish results.

**Disclaimer** The authors alone are responsible for the views expressed in this article and they do not necessarily represent the views, decisions or policies of the institutions with which they are affiliated.

**Competing interests** None declared.

**Patient and public involvement** Patients and/or the public were not involved in the design, or conduct, or reporting, or dissemination plans of this research.

**Patient consent for publication** Not applicable.

**Ethics approval** Ethical approval for data collection was obtained by the DHS Programme and implementing partners at time of survey, and participants provided informed consent to participate in the DHS survey at the time of data collection. This secondary data analysis using publicly available deidentified data was reviewed and approved by the National Health Sciences Research Committee of Malawi (#20220106).

**Provenance and peer review** Not commissioned; externally peer reviewed.

**Data availability statement** Data are available upon reasonable request. Data for this study are publicly available upon request through the DHS Programme. The data are owned by the Government of Malawi and are archived and managed by the DHS Programme. Interested researchers must register at https://dhsprogram.com/data/ and can then request permission to download the SPA and DHS datasets. Analytical code for results presented here is available upon reasonable request from the corresponding author.

**ORCID iDs**
Nicole E Johns http://orcid.org/0000-0003-4513-4582
Ahmad Reza Hosseinpoor http://orcid.org/0000-0001-7322-672X
Mike Chisema http://orcid.org/0000-0002-9705-3699
M Carolina Danovaro-Holliday http://orcid.org/0000-0001-7324-9198
Katherine Kirkby http://orcid.org/0000-0002-6881-409X
Anne Schlotheuber http://orcid.org/0000-0001-6393-4095
Messeret Shibeshi http://orcid.org/0000-0002-0403-5505
Samir V Sodha http://orcid.org/0000-0003-3695-1046
Boston Zimba http://orcid.org/0000-0001-6420-0341

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
