## [Reviewer comments · BMJ Open]

ARTICLE DETAILS

TITLE (PROVISIONAL)	The association between childhood immunization coverage and proximity to health facilities in rural settings: A cross-sectional analysis of Service Provision Assessment 2013-14 facility data and Demographic and Health Survey 2015-16 individual data in Malawi
AUTHORS	Johns, Nicole; Hosseinpoor, Ahmad Reza; Chisema, Mike; Danovaro-Holliday, M Carolina; Kirkby, Katherine; Schlottheuber, Anne; Shibeshi, Messeret; Sodha, Samir V.; Zimba, Boston

VERSION 1 – REVIEW

REVIEWER	Dadari, Ibrahim University of South Florida I declare no competing interest but alluding to the fact that I know and work with some of the authors.
REVIEW RETURNED	01-Mar-2022

GENERAL COMMENTS	 • Title: could be revised to be more straightforward and could read “The association between childhood immunization coverage and proximity to health facilities in rural areas: A cross-sectional analysis of service provision assessment (SPA) SPA 2013-14 facility and demographic & health survey (DHS) 2015-16 individual data in Malawi.” • Abstract: The statement “Remote rural populations are increasingly recognized as a priority group for immunization service equity.” could be paraphrased. There is evidence in the literature showing that remote rural populations have been prioritized for quite some time now. It may also be good to reference the ERG discussion paper on remote rural populations. • Page 6 Line 32: 5km radius vs 8km radius Malawi policy on proximity to public health facilities. It will be great to provide some clarity and mention that the 8km distance policy of the Malawi MOH was also used/factored in the analysis. • The word Inequalities were used interchangeably with inequities throughout the text? • Use of the word vaccine-providing?? Could a better adjective be used such as vaccination sites or “facility offering vaccination services” as mentioned within the paper in some places • Page 7 Line 44: facility geolocation – pls mention source of data, is it DHS? • Page 7 Lines 44 – 48: phrase needs to be simplified for better comprehension by the reader. Also, good to highlight how the Malawi MOH 8KM radius was incorporated into the analysis. • Page 9 Lines 7 – 9: should be captured in the methodology modifying to proximity radius of 8km as recommended by Malawi DOH • Page 15 Lines 26 – 29: “Conversely, the null findings for 3-dose
--

	OPV and 3-dose pentavalent vaccine add evidence to the success of the Malawi EPI in ensuring access to these vaccines among rural populations more broadly." Please cite references alluding to the successes by Malawi in ensuring access to vaccines for rural populations
--	--

REVIEWER	Shenton, Luke University of Michigan School of Public Health, Epidemiology
REVIEW RETURNED	15-Mar-2022

GENERAL COMMENTS	Overall well written and has some interesting contributions to the literature. Please see specific suggestions below. My largest questions revolve around expanding the discussion section to show how your results can be used in a practical sense. Abstract  - Objective section doesn't currently have a clear statement of what you hope to add to the literature that isn't already there - Unclear what you mean by composite vaccine indicator in outcome measures Strengths and Limitations  - Do you mean first to study this using national data in Malawi? It is not the first study globally to do this. Please clarify Introduction  - Overall you cover important background, make the need for your study apparent, and state your objectives. - In some places you talk about the study being about children 12-23 months. In other places you talk about the study being about children under 1. Are you looking at children 12-23 months but only counting vaccines they received prior to 12 months. Please clarify. Methods  - Clear methods Results  - Some of the numbers in your text don't seem to match the tables. For instance you say 61% were within 5 km of vaccine providing facility. However table says 36.3% were within 5 km. Please carefully check your text and tables and make sure the numbers are consistent - When discussing by type of facility what is the comparison group. It seems that vaccine odds were higher in government and CHAM facilities. So is the comparison in relation to facilities that aren't in either of these categories. Please specify in text Discussion  - I think some more information on how vaccine outreach in Malawi works could be helpful. You acknowledge in your limitations that outreach isn't included in analysis and in methods that the database doesn't included mobile sites. So some additional discussion of how vaccine outreach in the country works could allow readers to better understand the impact this has on your results. Does Malawi have robust outreach programs or are they limited in nature. - Did you consider looking at increasing distances from vaccine sites. 8km is only marginally further than 5km. I wonder if you'd see decreasing vaccine with increasing distance. you show that 61% aren't within 5 km, what about 10 km 30 km, etc. Might be out of the scope of your paper but could be interesting to see. It could also add
--

	a lot to the usefulness of your analysis for planning purposes. For instance if is there a distance at which vaccine rates start to dramatically drop (how to decide where to put new facilities). I'm especially intrigued by this as multiple vaccines showed no difference at 5km. - I would like to see more discussion surround how this information can be used. You conclude that remote rural populations have been identified as a key target for improving immunization equity. You say that efforts should target remote communities far from health facilities. However, you didn't actually look at the impact of efforts in far communities, just whether they are close to a center or not. So what could be said for planning more outreach activities versus building new facilities?
--	--

REVIEWER	Francis, Mark Tampere University, Epidemiology
REVIEW RETURNED	16-Mar-2022

GENERAL COMMENTS	It was a pleasure to review this well-written paper. The study investigated the association between proximity to health facilities and a range of childhood vaccination indicators in remote rural populations utilizing publicly-available data from Malawi's Service Provision Assessment and Demographic and Health Surveys. Positive associations were observed between proximity to vaccination-providing facilities (within 5 km) and the uptake of rotavirus and measles vaccinations, regardless of the source of vaccination history. The introduction and rationale for the study, the methods, findings of the primary and secondary analyses, and discussion and conclusions drawn are well presented. I have minor comments for the authors' consideration attached below. Abstract: 1. Pg.3, lines 10-11: Consider rewording "analyses examining both health facility and individual data offer insight into immunization equity" to "analyses examining both health facility and individual data can/may offer insight into immunization equity." Introduction: 2. Pg.5, lines 11-13: It is important to define "remote rural populations" to begin with, as this term is frequently used throughout the manuscript. 3. Pg.5, lines 16-18: "Gavi's most recent strategy centers leaving no one behind..." This statement is not clear; please consider rewriting it. 4. Pg.6, lines 11-13: Are there examples of these "remote" populations in Malawi? Methods: 5. Pg.9, lines 3-6: The construction of the "multinomial models" is unclear. Please describe in greater detail. Results: 6. Pg.10, line 37: It would be helpful for the reader to include more detail, perhaps in a flowchart, on how the 2740 children were obtained from the DHS dataset. 7. Pg.10, lines 38-40: That 64% and 61% of children were within 5km of any health facility and vaccine-providing facility is not very apparent in Table 3. Please check. 8. Pg.11, lines 3-44: While Table 3 is well made, it would be helpful to have other socio-demographic information on the children or their
---

	families. 9. Pg.14, line 4-5: This statement "The majority of children in rural areas (80%, 95% CI 78%-82%) had a vaccination card which was seen" is repeated. Please delete Discussion and conclusions: 10. Pg.14, lines 50-53: Do the authors imply that the positive associations between proximity of children to a health facility providing vaccination services and rotavirus and measles vaccination was responsible for the associations with all recommended vaccines and zero-dose pentavalent vaccinations? 11. Pg.15, lines 9-12: Have previous definitions of remote populations concerning childhood vaccinations not considered a child's proximity to health facilities? If so, this needs to be made more evident. 12. Pg.15, lines 15-18: Could it also be that healthcare workers more frequently visit and remind parents about due vaccines if the health facility is more proximal to specific rural areas? 13. Pg.16, lines 1-26: Why was the association between the children's proximity to health facilities and timing of individual vaccine doses not investigated or discussed? It would have been interesting to see if children who lived closer to health facilities had more timely doses than those living further away. 14. Pg.16, lines 1-26: Was the 2015-16 DHS dataset the most recent dataset available for Malawi? If not, was the 2015-16 DHS dataset analyzed because it was the closest in time to the SPA facility surveys (2013-14)? I am not sure that this has been mentioned clearly in the manuscript.
--	--

REVIEWER	Geremew , Shewayiref Ethiopian Public Health Institute
REVIEW RETURNED	17-Mar-2022

GENERAL COMMENTS	Title: The association between childhood immunization coverage and proximity to health facilities in rural areas: A cross-sectional analysis of SPA 2013-14 facility and DHS 2015-16 individual data in Malawi. It is better to be "A Further Analysis of" Instead of "A cross-sectional analysis ...". Instead of "...between...", "...of...". Instead of "...areas...", "...settings ..." Page 6; Line 46: identify the sources of the data with the time point the data collected. Facility data come from SPA 2013-14 facility and DHS 2015-16 individual data in Malawi..... Page 6; Line 47: "census".... Do you mean "sample"? as SPA is Service Provision Assessment survey not a census. Page 7: Line 34: you have stated that "Several changes to the EPI have gone into effect since 15". However, the DHS data were collected in the year 2015-16. Is there no information have been missed, since there is an overlap between the changes in the EPI and the data were collected? Page 7; Line 45-47 and Page 9; line 43: There is no urban households included in the study. Why? Include the inclusion and exclusion criteria. Page 8; line 13: DHS is a community-based survey not an individual survey. Page 8; line 13: specify the methodology you used for the integration/triangulation of facility survey (SPA) and community-based survey (DHS)? Most of the outreach vaccination services are not
---

	considered/registered under the facility data repository due to nature of campaign. Because of this the facility services are under estimated. How you considered it? Page 8; line 9-11: (2%) reported charging vaccination-specific fees + 99% of facilities offering vaccines offered full EPI = 101% health facilities. Please check here, the numerical figures. Page 10; line 25-26: Do you conducted a survey or used the data from SPA and DHS? If you conducted a survey before 7 years ago, it was old data and too late. If you use secondary data from SPA and DHS, no need of informed consent from participants, as the data from DHS is publicly free to use for further analysis. Please clearly state this part. Page 11; line 48: instead of "...two-thirds(76.8%)..." it is better to say "...three-fourth(76.8%)..." Page 12; line 20: "...receiving all basic vaccines (AOR 1.28, 95% CI 0.99-1.65, p=0.057) ..." is not a significant value. Why you want to report it as having marginally greater odds? Page 12, line 23: "...zero-dose for pentavalent (AOR 0.53, 95% CI 0.28-1.01, p=0.052)..." is not significant value. Why you want to report it as having somewhat lower odds? Methodology part: There is no a clearly defined method on how the two data sources are integrated, taking in to mind that there is a difference in the time point of the data were collected? Under the conclusion part: The study used the data 6 or 7 years back from now. We know that the things are changing fast, especially due to the SDG agenda of 2030, the health sector doing on the reducing of maternal and child mortality and morbidity by services like immunization/vaccination for children. Therefore, the things were in 6 years back are not similar or nearly similar with now. So, how much your evidences can be used for public health policy-makers and officials rather than having and designing a methodology to do it?
--	---

VERSION 1 – AUTHOR RESPONSE

Reviewer: 1

1. Title: could be revised to be more straightforward and could read "The association between childhood immunization coverage and proximity to health facilities in rural areas: A cross-sectional analysis of service provision assessment (SPA) SPA 2013-14 facility and demographic & health survey (DHS) 2015-16 individual data in Malawi."

Based on this suggestion and feedback from reviewer 4, we have revised the title to read "The association between childhood immunization coverage and proximity to health facilities in rural settings: A cross-sectional analysis of Service Provision Assessment 2013-14 facility data and Demographic and Health Survey 2015-16 individual data in Malawi"

2. Abstract: The statement "Remote rural populations are increasingly recognized as a priority group for immunization service equity." could be paraphrased. There is evidence in the literature showing that remote rural populations have been prioritized for quite some time now. It may also be good to reference the ERG discussion paper on remote rural populations.

This is a good point; we have removed the qualifier and revised this statement to read "Remote rural populations are recognized...". We reference the ERG discussion paper in the introduction; it was instrumental in designing and framing our study.

3. Page 6 Line 32: 5km radius vs 8km radius Malawi policy on proximity to public health facilities. It will be great to provide some clarity and mention that the 8km distance policy of the Malawi MOH was also used/factored in the analysis.

For clarity, we have elected to explain use of both 5km and 8km radius in the methods/results sections of the manuscript, rather than introduce the additional analyses in the introduction. We maintain the focus on 5km given our initial pre-specified study hypothesis.

4. The word Inequalities were used interchangeably with inequities throughout the text?

Thank you for bringing this to our attention; we have standardized throughout the text to use 'inequalities' whenever discussing measured differences, and 'inequities' only when discussing theory and policy framing.

5. Use of the word vaccine-providing?? Could a better adjective be used such as vaccination sites or "facility offering vaccination services" as mentioned within the paper in some places.

We appreciate that this is a non-standard term and not as clear as 'facility offering vaccination services'. However, the repetitive use of the phrase (we currently use 'vaccine-providing' 40 times in text) makes the longer phrase become a bit overly wordy. We would therefore prefer to retain the phrase 'vaccine-providing' for simplicity.

6. Page 7 Line 44: facility geolocation – pls mention source of data, is it DHS?

Facility location GPS was collected as part of the SPA surveys; this is now noted in text.

7. Page 7 Lines 44 – 48: phrase needs to be simplified for better comprehension by the reader. Also, good to highlight how the Malawi MOH 8KM radius was incorporated into the analysis.

We have simplified and clarified this explanation, and added mention of the 8km analyses, as follows: "Therefore, we defined proximity to a vaccine-providing facility as children living in rural areas within 5km of a facility that reported regularly providing vaccination services on-site and/or via outreach. We also tested alternate buffer distances, including 8km based on Malawi Ministry of Health definitions (see sensitivity analysis description below)."

8. Page 9 Lines 7 – 9: should be captured in the methodology modifying to proximity radius of 8km as recommended by Malawi DOH

Additional mention of this distance has been added to the methods section as noted above (see comment 7).

9. Page 15 Lines 26 – 29: "Conversely, the null findings for 3-dose OPV and 3-dose pentavalent vaccine add evidence to the success of the Malawi EPI in ensuring access to these vaccines among rural populations more broadly." Please cite references alluding to the successes by Malawi in ensuring access to vaccines for rural populations

We have added reference to the most recent published Malawi EPI document, which defines and indicates geographic equity based on 100% of districts having $\geq 80\%$ coverage of DTP3.

Reviewer: 2

Overall well written and has some interesting contributions to the literature. Please see specific suggestions below. My largest questions revolve around expanding the discussion section to show how your results can be used in a practical sense.

We thank the reviewer for these kind words and helpful suggestions below.

Abstract

1. Objective section doesn't currently have a clear statement of what you hope to add to the literature that isn't already there

We have added the following statement to clarify our intended addition to the literature: "We aimed to link facility and individual data to examine the relationship between distance to services and immunization coverage empirically, specifically using a rural population."

2. Unclear what you mean by composite vaccine indicator in outcome measures

We have added clarification that the composite indicators are "receipt of basic vaccines and receipt of all recommended vaccines".

Strengths and Limitations

3. Do you mean first to study this using national data in Malawi? It is not the first study globally to do this. Please clarify

Due to major revision of the Strengths and Limitations section as guided by the Editors, this statement has been removed.

Introduction

4. Overall you cover important background, make the need for your study apparent, and state your objectives.

Thank you!

5. In some places you talk about the study being about children 12-23 months. In other places you talk about the study being about children under 1. Are you looking at children 12-23 months but only counting vaccines they received prior to 12 months. Please clarify.

We examined children 12-23 months, and vaccines received at any point prior to survey (e.g. included those vaccines received after age 1). This has been clarified in methods text as follows:

“Immunization coverage was defined as the proportion of children aged 12–23 months who received the indicated vaccine dose by the time of survey” and “... vaccines received within the recommended time frame and any time after were included.” In the results, we clarify that the vast majority of vaccines were received before age 1, with the exception of measles vaccine: “Among children with vaccination date information available, all vaccines except measles had 99% or greater receipt within the first year of life, while 93.4% of measles first doses were received before age one (results not shown).”

Methods

6. Clear methods

Thank you!

Results

7. Some of the numbers in your text don't seem to match the tables. For instance you say 61% were within 5 km of vaccine providing facility. However table says 36.3% were within 5 km. Please carefully check your text and tables and make sure the numbers are consistent

Thank you for catching this error. We have revised the error in the table and confirmed all other presented numbers in tables and text.

8. When discussing by type of facility what is the comparison group. It seems that vaccine odds were higher in government and CHAM facilities. So is the comparison in relation to facilities that aren't in either of these categories. Please specify in text

As children could be proximal to multiple facilities, and there is not a clear ordering/hierarchy of facility ownership, we analyzed facility ownership as a set of indicator variables (proximal to a government facility offering immunization, yes/no; proximal to a Christian Health Association of Malawi (CHAM) facility offering immunization, yes/no; etc). Therefore the comparison as noted is government vs all others (including CHAM), for example. We have clarified this variable construction in the methods text as follows: “We captured facility managing authority via a set of four indicators, separately assessing whether a child was proximal to a vaccine-providing facility managed by the government, CHAM, private for-profit, and/or NGO.”

Discussion

9. I think some more information on how vaccine outreach in Malawi works could be helpful. You acknowledge in your limitations that outreach isn't included in analysis and in methods that the database doesn't included mobile sites. So some additional discussion of how vaccine outreach in the country works could allow readers to better understand the impact this has on your results. Does Malawi have robust outreach programs or are they limited in nature.

Thank you for highlighting that this could use further discussion. Outreach clinics are very common in Malawi, with more than 5000 outreach clinics noted in most recent published materials. There is not published data regarding the frequency of outreach vs in-facility receipt of immunization services, but use of outreach clinics is understood to be widespread. Ultimately, having data only on location of the nearest static clinic therefore likely biases our results towards the null, as many children likely received immunization services at outreach locations. We have noted this in the limitations section as

follows: “Additionally, outreach services are widely used in this population, with more than 5000 fixed and mobile outreach clinics throughout the country; the observed association of immunization coverage with distance to static clinics likely underestimates the true strength of association with distance to location where vaccination was actually received.”

10. Did you consider looking at increasing distances from vaccine sites. 8km is only marginally further than 5km. I wonder if you'd see decreasing vaccine with increasing distance. you show that 61% aren't within 5 km, what about 10 km 30 km, etc. Might be out of the scope of your paper but could be interesting to see. It could also add a lot to the usefulness of your analysis for planning purposes. For instance if is there a distance at which vaccine rates start to dramatically drop (how to decide where to put new facilities). I'm especially intrigued by this as multiple vaccines showed no difference at 5km.

This is a good point, and raises the potential for future work. We did in fact examine a 10km radius; we did not examine any greater distances because 96% of children resided within 10km of a facility providing vaccination services or outreach. We do note a somewhat stronger strength of association for measles at the 10km distance (AOR 2.05, 95%CI 1.03-4.09), and the association with rotavirus is lost with this threshold (AOR 0.99, 95% CI 0.47-2.08). Ultimately, we chose to present 5km and 8km findings based on the data definitions used by DHS (e.g. their suggested matching based on 5km radius, and 5km offsets of cluster centroids) and the Malawi Ministry of Health policy (stating that all Malawians should live within 8km of a facility), as well as the fact that very low vehicle ownership necessitates a shorter, walk-able distance to ensure access. Details regarding low vehicle ownership have been added to the introduction to add further clarity as to why the shorter 5km and 8km distances were used. “The 8km policy is intended to ensure “reasonable walking distance” to healthcare, which is particularly relevant given low vehicle ownership (only 4% of households in rural areas had a car, truck, motorcycle or scooter as of the 2015-16 DHS).”

11. I would like to see more discussion surround how this information can be used. You conclude that remote rural populations have been identified as a key target for improving immunization equity. You say that efforts should target remote communities far from health facilities. However, you didn't actually look at the impact of efforts in far communities, just whether they are close to a center or not. So what could be said for planning more outreach activities versus building new facilities?

This point is well taken. The primary programmatic implication of these findings is that even the crude measure of straight-line distance is a meaningful correlate of coverage, and therefore a useful (and simple) tool for policy-makers and officials to identify potentially under-vaccinated populations. As analyses could not separate vaccines received in facilities from those received during outreach activities (nor where those outreach activities occurred), we do not have sufficient data to draw conclusions on the relative strengths of static facilities vs outreach efforts. We have made revisions to the limitations and conclusions to better address these points.

Reviewer: 3

It was a pleasure to review this well-written paper. The study investigated the association between proximity to health facilities and a range of childhood vaccination indicators in remote rural populations utilizing publicly-available data from Malawi's Service Provision Assessment and Demographic and Health Surveys. Positive associations were observed between proximity to vaccination-providing facilities (within 5 km) and the uptake of rotavirus and measles vaccinations, regardless of the source of vaccination history. The introduction and rationale for the study, the methods, findings of the primary and secondary analyses, and discussion and conclusions drawn are well presented. I have minor comments for the authors' consideration attached below.

We thank the reviewer for their kind words and helpful feedback below.

Abstract:

1. Pg.3, lines 10-11: Consider rewording "analyses examining both health facility and individual data offer insight into immunization equity" to "analyses examining both health facility and individual data can/may offer insight into immunization equity."

Due to abstract word count limits, we have removed this sentence entirely.

Introduction:

2. Pg.5, lines 11-13: It is important to define "remote rural populations" to begin with, as this term is frequently used throughout the manuscript.

We have clarified that remote rural populations are "those living furthest from population centers". There is not a strict/uniform specific distance which is used to define remote rural across contexts, so we have worded it to encompass the range of definitions used.

3. Pg.5, lines 16-18: "Gavi's most recent strategy centers leaving no one behind..." This statement is not clear; please consider rewriting it.

We have rephrased this to read: "Similarly, Gavi's most recent strategy also centers the theme of leaving no one behind..."

4. Pg.6, lines 11-13: Are there examples of these "remote" populations in Malawi?

We have added mention of data from the Malawi DHS, examining the range of vaccine coverage across rural strata. We observe a large difference across these rural groups, highlighting within-rural inequalities. The additional text in the introduction is: "Data from the 2015-16 Malawi DHS confirms these within-rural inequalities: coverage of basic vaccines varied by more than 25 percentage points across rural strata, from 65% (95% CI 53%-75%) to 91% (95% CI 84%-95%)."

Methods:

5. Pg.9, lines 3-6: The construction of the "multinomial models" is unclear. Please describe in greater detail.

Additional detail has been added to clarify as follows: "For each vaccine, we defined a 3-level outcome as: not received (referent group), received and verified by vaccine card, and received as noted via mother's recall."

Results:

6. Pg.10, line 37: It would be helpful for the reader to include more detail, perhaps in a flowchart, on how the 2740 children were obtained from the DHS dataset.

Inclusion criteria have been added in text to clarify how the analytic sample was determined: "All living children aged 12-23 months residing in rural areas were included."

7. Pg.10, lines 38-40: That 64% and 61% of children were within 5km of any health facility and vaccine-providing facility is not very apparent in Table 3. Please check.

Thank you for flagging this typo. We have modified the table and confirmed other numbers in text and tables throughout.

8. Pg.11, lines 3-44: While Table 3 is well made, it would be helpful to have other socio-demographic information on the children or their families.

Additional descriptive statistics have been added in the new Appendix Table 1.

9. Pg.14, line 4-5: This statement "The majority of children in rural areas (80%, 95% CI 78%-82%) had a vaccination card which was seen" is repeated. Please delete

The repeated phrase has been removed, thank you for noting that error.

Discussion and conclusions:

10. Pg.14, lines 50-53: Do the authors imply that the positive associations between proximity of children to a health facility providing vaccination services and rotavirus and measles vaccination was responsible for the associations with all recommended vaccines and zero-dose pentavalent vaccinations?

We imply that the association with 'all recommended vaccines' is related to the association with measles and rotavirus vaccination, as those are components of the 'all recommended vaccine' indicator; we do not imply such an association with zero-dose. The text has been edited to clarify this as follows: "...was associated with increased likelihood of rotavirus and measles vaccine receipt (and therefore receipt of all recommended vaccines, as rotavirus and measles are part of this composite indicator), as well as with decreased likelihood of zero-dose pentavalent vaccination."

11. Pg.15, lines 9-12: Have previous definitions of remote populations concerning childhood vaccinations not considered a child's proximity to health facilities? If so, this needs to be made more evident.

Previous research has examined child proximity to health facilities, but analyses have either examined measured distances for smaller populations (e.g. not nationally representative, as we have examined here) or have assessed distance to care in nationally representative samples using questions about perceived distance to care (e.g. health facility is 'too far'). However, additional examination of the literature based on this comment did return one additional study from Nigeria (Sato 2020) examining facility census data and DHS data to assess distance to care in a nationally representative sample; mention of this study has been added to the introduction.

12. Pg.15, lines 15-18: Could it also be that healthcare workers more frequently visit and remind parents about due vaccines if the health facility is more proximal to specific rural areas?

This may in fact be the case, but we don't have data to test this hypothesis. However, we would not expect this to result in differential results by vaccine type.

13. Pg.16, lines 1-26: Why was the association between the children's proximity to health facilities and timing of individual vaccine doses not investigated or discussed? It would have been interesting to see if children who lived closer to health facilities had more timely doses than those living further away.

We intended to examine receipt within the first year of life for vaccines. We found surprisingly high "timely" (e.g. by age one) receipt for all vaccines except measles, rendering further examination of timeliness not analytically feasible. For children with vaccination date available, all except measles had >99% receipt within the first year of life. Measles vaccine was received by 93.4% of children by age 1; the lower receipt of measles by age 1 is unsurprising given that its recommended administration age is much later (9 months). We have added note of this information to the methods section as follows: "Among children with vaccination date information available, all vaccines except measles had 99% or greater receipt within the first year of life, while 93.4% of measles first doses were received before age one (results not shown)." We have also noted that the intended sensitivity analysis was not possible in the limitations. Further examination of exact timing of vaccine receipt, and whether this represents uncorrected missed opportunities for vaccination in the second year of life, is beyond the scope of the current analyses.

14. Pg.16, lines 1-26: Was the 2015-16 DHS dataset the most recent dataset available for Malawi? If not, was the 2015-16 DHS dataset analyzed because it was the closest in time to the SPA facility surveys (2013-14)? I am not sure that this has been mentioned clearly in the manuscript.

Yes, the 2015-16 DHS dataset is the most recently available, and the 2013-14 survey was the most recently available SPA dataset, which is why these datasets were used. This has been clarified in the methods and limitations sections, with the added limitation that the data are now 6+ years old. Methods text: "We used data from the Malawi 2015-16 DHS, the most recent available DHS survey in the country, to determine immunization coverage among children aged 12-23 months residing in rural areas."; Limitations text: "... while the most recent available surveys were used, at time of publication these data are now 6-9 years old; additional research using more recent data will add insight into current realities."

Reviewer: 4

1. Title: "The association between childhood immunization coverage and proximity to health facilities in rural areas: A cross-sectional analysis of SPA 2013-14 facility and DHS 2015-16 individual data in Malawi." It is better to be "A Further Analysis of" Instead of "A cross-sectional analysis ...". Instead of "...between...", "...of...". Instead of "...areas...", "...settings ..."

Based on this suggestion and feedback from Reviewer 1, we have revised the title to read "The association between childhood immunization coverage and proximity to health facilities in rural settings: A cross-sectional analysis of Service Provision Assessment 2013-14 facility data and Demographic and Health Survey 2015-16 individual data in Malawi"

1. Page 6; Line 46: identify the sources of the data with the time point the data collected. Facility data come from SPA 2013-14 facility and DHS 2015-16 individual data in Malawi.....

The data sources and survey years are now mentioned and referenced in the 'Data sources' section of the Methods.

1. Page 6; Line 47: "census".... Do you mean "sample"? as SPA is Service Provision Assessment survey not a census.

Interestingly, the Malawi 2013-14 SPA was one of the few SPA surveys conducted as a full census. That is part of why it was selected for analysis, as all facilities would be captured by the data.

1. Page 7; Line 34: you have stated that "Several changes to the EPI have gone into effect since 15". However, the DHS data were collected in the year 2015-16. Is there no information have been missed, since there is an overlap between the changes in the EPI and the data were collected?

This is correct; we have edited the footnote of Table 1 to clarify: "Though this schedule reflects the EPI schedule at time of data analyzed here, several changes to the EPI have gone into effect since 2015". We elect to maintain the footnote given that readers may be interested to know the current EPI schedule in the study area.

1. Page 7; Line 45-47 and Page 9; line 43: There is no urban households included in the study. Why? Include the inclusion and exclusion criteria.

We have *a priori* limited all analyses to rural populations, and have made this clearer in text throughout. We have modified text in the introduction, methods, and results to clarify that the study sample is children aged 12-23 months living in rural areas. The inclusion criteria are also clarified in the results section as follows "All living children aged 12-23 months residing in rural areas were included."

1. Page 8; line 13: DHS is a community-based survey not an individual survey.

This text has been edited to "survey of individuals" for clarity.

1. Page 8; line 13: specify the methodology you used for the integration/triangulation of facility survey (SPA) and community-based survey (DHS)?

This methodology is described in the 'Geographic data linkage' section, which has now been reworded for clarity: "Facility geo-location, provided as part of SPA data, was matched to individual data using the DHS-suggested technique of a Euclidian (straight-line) distance buffer around each DHS cluster centroid. This buffer is defined as 5km in rural areas; this corresponds with the DHS offset of cluster GPS locations by up to 5km in rural areas. Therefore, we defined proximity to a vaccine-providing facility as children living in rural areas within 5km of a facility that reported regularly providing vaccination services on-site and/or via outreach".

1. Most of the outreach vaccination services are not considered/registered under the facility data repository due to nature of campaign. Because of this the facility services are under estimated. How you considered it?

This point is well taken; please see response to Reviewer 2, point 9 for our response.

1. Page 8; line 9-11: (2%) reported charging vaccination-specific fees + 99% of facilities offering vaccines offered full EPI = 101% health facilities. Please check here, the numerical figures.

These are in fact two separate statistics which were poorly worded together; we have clarified this text by separating these into two statements: "Indicators of vaccination fees were not used because very few facilities (2%) reported charging vaccination-specific fees. Indicators of full EPI provision were not used because most facilities offering any vaccination services offered all EPI vaccines (99% of facilities offering vaccines offered full EPI)."

1. Page 10; line 25-26: Do you conducted a survey or used the data from SPA and DHS? If you conducted a survey before 7 years ago, it was old data and too late. If you use secondary data from SPA and DHS, no need of informed consent from participants, as the data from DHS is publicly free to use for further analysis. Please clearly state this part.

This point is well taken, but the journal requires that we state informed consent procedures for the original data collection [*editors, please correct if we have misunderstood*]. We have added clarification that the informed consent was for DHS survey participation, and that only the secondary data analysis (this study) was reviewed by the NHRSCM, not the DHS surveys: "Ethical approval for data collection was obtained by the DHS Program and implementing partners at time of survey, and participants provided informed consent to participate in the DHS survey at the time of data collection. This secondary data analysis using publicly-available de-identified data was reviewed and approved by the National Health Sciences Research Committee of Malawi (#20220106)."

1. Page 11; line 48: instead of "...two-thirds(76.8%)..." it is better to say "...three-fourth(76.8%)..."

Thank you for noting this error. We have revised to three-fourths, as suggested.

1. Page 12; line 20: ".....receiving all basic vaccines (AOR 1.28, 95% CI 0.99-1.65, p=0.057) ..." is not a significant value. Why you want to report it as having marginally greater odds?

You are correct in noting that this is not strictly statistically significant at the threshold $p < 0.05$. However, we wish to avoid overreliance on p-values to draw conclusions about our hypotheses, and present these findings in the context of other immunization effect sizes and strengths of association [see, for example, Wasserstein, Ronald L., and Nicole A. Lazar. "The ASA statement on p-values: context, process, and purpose." *The American Statistician* 70.2 (2016): 129-133]. The similar effect sizes and directions of association add to the collective evidence our analyses present, which is why we include the 'marginal' findings as well, being careful to note them as marginal and not strictly statistically significant at $p < 0.05$.

1. Page 12, line 23: "...zero-dose for pentavalent (AOR 0.53, 95% CI 0.28-1.01, p=0.052)..." is not significant value. Why you want to report it as having somewhat lower odds?

Please see response to comment 12, above.

1. Methodology part - There is no a clearly defined method on how the two data sources are integrated, taking in to mind that there is a difference in the time point of the data were collected?

Data were linked based on the GPS coordinate of the facility and the GPS coordinate of the DHS cluster centroid (as in DHS, individual household GPS is not provided, only cluster centroid). Though there is a gap in time between surveys, there is retroactive reporting of immunizations which were likely received within the time range of the SPA survey. We have added further detail to the 'Geographic data linkage' section in the methods to clarify how data were linked; in the measures section we have added clarification of the time gap as follows: "Despite a 2-3 year gap between the survey of facilities (2013-14) and survey of individuals (2015-16, children born October 2013-February 2015), retroactive reporting of immunization for children 12-23 months reflects immunization receipt close to the time of facility survey, and provision of any vaccination services is likely to have remained constant over that time." We have also noted the tie period mismatch between the SPA and DHS in the limitations.

1. Under the conclusion part - The study used the data 6 or 7 years back from now. We know that the things are changing fast, especially due to the SDG agenda of 2030, the health sector doing on the reducing of maternal and child mortality and morbidity by services like immunization/vaccination for children. Therefore, the things were in 6 years back are not similar or nearly similar with now. So, how much your evidences can be used for public health policy-makers and officials rather than having and designing a methodology to do it?

This is a good point, and the age of the data is ultimately a limitation of the study. We have therefore added the following to the limitations section: "Fourth, while the most recent available surveys were used, at time of publication these data are now 6-9 years old; additional research using more recent data will add insight into current realities." Despite this, the broad finding that coverage was negatively associated with distance from a facility, even using a very crude measure of distance which does not take into account actual household location nor geographic/seasonal/other measures of access, provides an extremely simple and straightforward way for policy-makers and officials to identify and target populations which may be under-vaccinated (e.g. anyone outside of a 5km radius of a facility offering services). This manuscript also serves to demonstrate feasibility of linking facility and individual-level survey data for distance analyses, which is not sensitive to the age of the data.

VERSION 2 – REVIEW

REVIEWER	Dadari, Ibrahim University of South Florida
REVIEW RETURNED	23-Apr-2022
GENERAL COMMENTS	Great work and the paper came out stronger with the revisions. I think this paper will definitely contribute to the equity discussion and prioritization. All the best.
REVIEWER	Shenton, Luke University of Michigan School of Public Health, Epidemiology
REVIEW RETURNED	05-May-2022
GENERAL COMMENTS	Introduction: clear and well written. Good idea of current policies and background on distance to facilities and immunization. Clear description of what your study adds that isn't already well studied. Methods: Clear, reproducible.

	Discussion: Overall, conclusions are clear and logically reached. However, I would like to see some more discussion on how this data can specifically be used to improve programs. As you mentioned the finding that distance to health facility affects vaccination isn't particularly novel. Making more ties to how this information can specifically be used in Malawi would be novel. You mention in methods that this analysis doesn't account for outreach programs. Based on your data it seems that outreach programs might be helping in keeping most vaccine rates high (w/ exception of MMR and rotavirus). So as far as expending on how this data is useful you could consider strategies such as needing more facilities vs. just boosting outreach for these specific vaccines. Did the data source you used touch on vaccine stock. Whether a center actually has the vaccines in stock when people are at the center. For limitations, any thoughts on the impact of SPA survey and other data being from different years? E.g. expansion of center prior to other data you used. Overall, well written paper and importance to literature is described. However, given that finding isn't particularly novel I have given some suggestions for making the papers contribution to literature a little clearer.
--	--

REVIEWER	Francis, Mark Tampere University, Epidemiology
REVIEW RETURNED	06-May-2022

GENERAL COMMENTS	Thank you for addressing my comments. The manuscript reads much better now.
---

REVIEWER	Geremew , Shewayiref Ethiopian Public Health Institute
REVIEW RETURNED	15-Apr-2022

GENERAL COMMENTS	For authors: Thank you for your very interested findings.
---

VERSION 2 – AUTHOR RESPONSE

Reviewer 1: Dr. Ibrahim Dadari, University of South Florida

Comments to the Author: Great work and the paper came out stronger with the revisions. I think this paper will definitely contribute to the equity discussion and prioritization. All the best.

Reviewer 3: Mr. Mark Francis, Tampere University, Tampere University

Comments to the Author: Thank you for addressing my comments. The manuscript reads much better now.

Reviewer 4: Mr. Shewayiref Geremew , Ethiopian Public Health Institute

Comments to the Author: For authors: Thank you for your very interested findings.

The authors thank reviewers 1, 3, & 4 for their initial feedback, which significantly strengthened the manuscript.

Reviewer 2: Mr. Luke Shenton, University of Michigan School of Public Health

Comments to the Author:

Introduction: clear and well written. Good idea of current policies and background on distance to

facilities and immunization. Clear description of what your study adds that isn't already well studied.
 Methods: Clear, reproducible.
 Discussion: Overall, conclusions are clear and logically reached.

1. However, I would like to see some more discussion on how this data can specifically be used to improve programs. As you mentioned the finding that distance to health facility affects vaccination isn't particularly novel. Making more ties to how this information can specifically be used in Malawi would be novel. You mention in methods that this analysis doesn't account for outreach programs. Based on your data it seems that outreach programs might be helping in keeping most vaccine rates high (w/ exception of MMR and rotavirus). So as far as expending on how this data is useful you could consider strategies such as needing more facilities vs. just boosting outreach for these specific vaccines.

We have added the following text to the discussion to provide more specific detail on how this data should inform programs/policy; while these suggestions are not necessarily novel, we do feel that the novel use of nationally-representative empirical data adds data-driven support to the need for rural-focused outreach efforts.

Practically, these additional efforts should include campaigns and outreach efforts, as these are less resource-intensive to implement than the construction, staffing, and maintenance of new facilities. These outreach efforts can be tailored to reach the most rural populations by inclusion of transportation considerations such as providing cars, motor bikes, and fuel, as well as supplies which can be carried long distances and be used in areas with limited infrastructure. Our findings also add further support to the stated goal of the Malawi Ministry of Health that all Malawians live within 8km of a health facility, and the construction of additional facilities should continue to prioritize those areas where people are furthest from care.

2. Did the data source you used touch on vaccine stock. Whether a center actually has the vaccines in stock when people are at the center.

Yes, vaccine stock was assessed by the SPA surveys. Though we examined current stock levels, we ultimately decided not to include current stock due to greater fluctuations in availability over time (relevant to your next piece of feedback regarding mismatched timing of surveys). At the facility level, non-expired vaccine was available and observed or reported among 74% (polio vaccine) to 81% (measles vaccine) of facilities that offered childhood immunization services in the country. At the child level, limited to the rural population analyzed in this paper, vaccine availability at a proximal facility offering any immunization services ranged from 80% (polio vaccine) to 87% (measles vaccine). To further examine the potential relationship between actual availability and immunization receipt, we conducted two additional sensitivity analyses: replicating our adjusted regression models with an indicator of current specific vaccine stock availability within 5km radius in place of an indicator of immunization service availability within 5km radius; and the same models limited to children within 5km of a facility offering immunization services (see table below). Findings were consistent in that current stock of measles and rotavirus vaccines were associated with greater likelihood of measles and rotavirus vaccination receipt; when limited to children who were within 5km of a facility offering immunization services, however, stock at time of SPA survey was not associated with coverage at time of DHS survey. This is likely due in part to changes in stock-outs over time and discordant timing of SPA and DHS surveys. We have added note that this additional analysis was conducted in methods and results sections, but to preserve manuscript length, and due to the fact that several sensitivity analyses are already included, we mention the results only briefly in text.

Adjusted odds of receipt of 6 vaccines among children 12-23mo living in rural areas

	Full population – facility offering immunization services within 5km vs not (Current analyses in paper, Table 4)	Full population – specific vaccine in stock within 5km vs not	Limited to children within 5km of a facility providing immunization services - specific vaccine in stock vs not
--	---	---	---

	AOR	95% CI	AOR	95% CI	AOR	95% CI
BCG	1.17	[0.63,2.20]	0.90	[0.48,1.70]	0.26	[0.05,1.39]
Rotavirus	1.63**	[1.13,2.33]	1.67**	[1.17,2.39]	1.55	[0.90,2.65]
Polio	1.08	[0.83,1.40]	1.17	[0.90,1.52]	1.29	[0.94,1.78]
Pentavalent	1.23	[0.84,1.82]	1.20	[0.81,1.77]	1.07	[0.58,1.97]
PCV13	1.07	[0.75,1.53]	1.17	[0.83,1.66]	1.43	[0.91,2.24]
Measles	1.62*	[1.11,2.37]	1.52*	[1.04,2.22]	1.08	[0.58,2.02]

Text added to methods & results is as follows:

[Methods] Fourth, as a final post-hoc analysis, we examined vaccine stock at time of SPA survey, considering a vaccine in stock if it was observed, or reported to be available, and was not expired. For each of the six examined vaccines, we replicated adjusted analyses of immunization coverage for the total sample using an indicator of the corresponding vaccine stock within 5km, and the same analyses limited to children within 5km of a vaccine-providing facility.

[Results] Current vaccine stock was positively and significantly associated with rotavirus (AOR 1.67, 95% CI 1.17-2.39) and measles (AOR 1.52, 95% CI 1.04-2.22) immunization coverage; these findings were consistent to those using indicators of immunization service availability generally (results not shown). When limited to children living within 5km of a vaccine-providing facility, vaccine stock was not associated with immunization coverage for any of the six examined vaccines.

3. For limitations, any thoughts on the impact of SPA survey and other data being from different years? E.g. expansion of center prior to other data you used.

We have considered this point, and discuss it both in methods and now in the limitations section. Retroactive reporting of immunization service receipt actually places time of immunization close to time of SPA survey, but we cannot say for certain that SPA data was accurate at the time of DHS-reported immunization receipt. Ultimately, the choice of the measure of immunization service availability generally is less likely to fluctuate over time than specific vaccine stock, for example, and we feel confident that the reality at time of SPA is at least a close approximation of reality at time of DHS-reported vaccine receipt. The following text has been added to limitations:

Furthermore, there was a 2-3 year gap between the survey of facilities (2013-14) and survey of individuals (2015-16, children born October 2013-February 2015), and facilities may have closed, opened, changed service offerings, or had fluctuations in vaccine stock over that time frame. However, retroactive reporting of immunization reflects immunization receipt close to the time of facility survey, and provision of any vaccination services is likely to have remained constant over that time. The lack of association between vaccine stock and immunization coverage may be due in part to the asynchronous surveys.

Overall, well written paper and importance to literature is described. However, given that finding isn't particularly novel I have given some suggestions for making the papers contribution to literature a little clearer.

We thank the reviewer for their thoughtful review and feedback to strengthen the manuscript.